# Evaluating the Effect of Different Stress Path Regimes on Borehole Deformation Using Convergence Measuring Device

**Jun Hyuk Heo [1,*], Noune Melkoumian [1] and Sam S. Hashemi [2]**

1   School of Civil, Environmental, and Mining Engineering, University of Adelaide, Adelaide, SA 5000, Australia
2   Department of Energy Resources Engineering, Stanford University, Stanford, CA 94305, USA
*   Correspondence: jun.heo@adelaide.edu.au

**Abstract:** A laboratory study was conducted to investigate the borehole deformation of poorly cemented sandstone rocks with Uniaxial Compressive Strength (UCS) less than 10 MPa under different stress path regimes by using a convergence measuring device (CMD). Synthetic thick-walled hollow cylinders (TWHCs) comprised of sand grains, Portland cement and water were prepared for this study. A series of mechanical tests including uniaxial and triaxial compression tests were performed to examine the physical properties of the artificial sandstones. A vertical displacement loading rate of 0.07 mm/min and confining pressure at a rate of 0.2 MPa/min were applied for the experiments. The CMD was deployed inside the TWHC specimen to measure the borehole deformation. Five different stress paths were applied to the specimens to investigate the effect of stress paths, and three different cement agent contents (10%, 12% and 14%) were considered to study the effect of cement content on the borehole failure. The effect of the cement content on the borehole failure was found to be more significant than the effect of change in stress path regimes.

**Keywords:** borehole deformation; convergence measuring device; thick-walled hollow cylinder; borehole monitoring

## 1. Introduction

Drilling boreholes and creating other cavities in the ground disturbs the local in situ stress field due to the stress concentration around the cavity. Under the stress concentration, weak bonding between the particles in poorly cemented formation may break when there is an inadequate cement agent present between the interfaces of sand grains, leading to inelastic damaged zone and borehole failure. Understanding the responses of poorly cemented formation is important for the prediction of failure modes and mechanical characteristics. Borehole convergence is a fundamental parameter in borehole stability analysis involving deformational calculations. In order to ensure long term stability of modern drilling operations, the acquisition of geotechnical information about encountered formation is critical for early detection of instability issues and minimizing borehole failure. While various convergence measuring devices such as caliper log, borehole extensometer, and convergence monitor are being used in the field, relatively few efforts have been made to accurately measure the borehole convergence in laboratory experiments for weak rocks under different stress path regimes. The cell of Bonnechere, the USBM gage and the CSIRO HI cell have been proposed for laboratory-based tests to measure the borehole deformation [1]. However, their application requires an extensive installation procedure and is limited to a specific borehole diameter size matching the size of the monitoring device, i.e., the diameter must be more than 30 mm. In the current study, a cost-effective, versatile and reliable convergence measuring device with a quick and easy installation procedure was developed based on the concept of the LDT proposed by Goto et al. [2] to evaluate the borehole deformation in thick-walled hollow cylinder specimens. These new CMDs have been calibrated and tested in laboratory conditions in order to verify their performance under different stress paths for poorly cemented sands.

Laboratory experiments involving the measurement of the diametric deformation of a borehole by placing a two-arm caliper inside a hollow cylinder were carried out by Tronvoll [3]. However, the effect of different stress paths was not investigated. Igarashi et al. [4] developed a borehole tangential deformation gage to measure the tangential deformation of the borehole wall. Papamichos et al. [5] performed laboratory experiments on hollow cylinder specimens of Red Wildmore sandstone where they used cantilever strain gage pairs to study the effect of the stress anisotropy on the borehole deformation. Bujok et al. [6] conducted experimental studies on the effects of the well convergence on the deformational shape of the borehole. Wu et al. [7] conducted polyaxial tests with a four-arm cantilever borehole gage placed inside the borehole within a block specimen to measure the borehole deformation. These tests involved a relatively stronger sandstone with a uniaxial compressive strength (UCS) greater than 10 MPa.

Laboratory experiments are found to be one of the most effective methods for studying the borehole failure mechanism. An experimental study on the effect of borehole inclination was conducted by Rawlings et al. [8] using both synthetic and natural weak sandstone specimens. Haimson and Song [9] observed a deep and narrow 'slot' shaped breakout in poorly cemented sandstone from a TWHC test. Papamichos et al. [5] studied the effect of anisotropic stress paths on weak natural sandstone. Younessi et al. [10] conducted laboratory tests on TWHC as well as true triaxial tests on synthetic sandstone specimens representing poorly cemented formations. Bujok et al. [6] studied the effect of borehole convergence using a specially made pressure vessel by applying confining pressure onto a wellbore model consisting of different layers of rock materials. Hashemi et al. [11] carried out TWHC tests on poorly cemented sands focusing on the effects of cement content, anisotropic stress paths and borehole sizes. Khormali et al. [12,13] conducted experimental studies on reservoir rocks. Li et al. [14] demonstrated that the effect of confining pressure in the range of 2 MPa ~ 13 MPa using hollow cylinder specimens on two different sandstones was nearly the same. Yan et al. [15] conducted a comparative study between unconsolidated and weakly consolidated sandstones and concluded that weakly consolidated sandstones tended to develop localised failure, whereas unconsolidated sandstones failed uniformly. Furthermore, Nouri et al. [16] analysed failure modes of sandstones, and Fu et al. [17] recently investigated the breakage process and the crack propagation paths of cubic specimens. However, limited laboratory-based experimental studies have been conducted to investigate the effect of borehole deformation using borehole convergence measuring sensors. This indicates a needs for the development of sensors that are capable of accurately measuring borehole convergence.

This study aims to investigate the application of the CMD on monitoring borehole deformation under different stress path regimes. A series of laboratory-based TWHC tests were designed and conducted under various stress paths. Both far-field and an element on the borehole wall were considered for applying different stress paths. The borehole behaviour was monitored by a video camera to the record borehole in order to determine the borehole failure along the borehole for the specimens. The results present a realistic understanding of the failure behaviour of poorly cemented sand formations under different stress paths.

## 2. Experimental Study

The Uniaxial Compressive Strength (UCS), triaxial and TWHC tests are the most widely used and versatile rock mechanics tests for determining the parameters required for borehole stability analysis and understanding the behaviour of a granular formation under different stress conditions. In this study, the borehole stability experiments were performed on TWHC specimens in a modified Hoek cell. During each test, the CMDs were deployed inside the borehole of the TWHC specimen for measuring the borehole diameter convergence, and a micro camera with the resolution of 225 pixels per inch (PPI) was placed inside the hollow platen of the Hoek cell for recording the borehole wall deformation during the testing and for its real-time visual monitoring. The TWHC specimen was

subjected to an axial stress along the borehole axis, and the confining stress was provided by a servo-controlled hydraulic device.

A number of laboratory facilities were used for conducting the tests. For applying vertical stress to the specimen, a servo-controlled axial loading system of 300 kN capacity with 0.1 N accuracy was used. For applying and maintaining the external confining pressure at a very low level, the Hoek cell was connected to an automatic hydraulic machine which had a relief valve and a pressure gage with an accuracy of 0.01 MPa. A 60-channel data acquisition system was connected to computers to monitor and record axial force, axial displacement, axial strain, lateral strain, convergence measurement and time into a storage device.

## 3. Lab Testing Program

The TWHC specimens were prepared in the laboratory under controlled conditions to manufacture poorly cemented sandstone specimens that resemble those of the natural rocks. Synthetic rock specimens are frequently used as an alternative to natural rocks for conducting laboratory-based borehole stability experiments [10,11]. The preparation of synthetic rocks involves mixing sand grains, a cementing agent and water. The mechanical properties of the prepared synthetic rock specimens depend on the individual components used in the mixture. A small variation in the initial components has a significant influence on the mechanical properties of the final specimens [10]. This indicates the importance of the careful selection of the components for the mixture. It has been observed that the mechanical properties of the synthetic rocks are primarily controlled by the cementation as the sand grains are bonded together by a cementing agent [18]. Liévano and Kanji [19] tested synthetic rock specimens made under controlled conditions and found that the mechanical properties of the synthetic specimens closely resemble those of the natural rocks.

Synthetic sandstone specimens which are close to the weak formations in terms of mechanical parameters such as the UCS are less than 10 MPa. For further experimental study of borehole convergence behaviour of this type of sandstone, other parameters such as high porosity and permeability were replicated. The synthetic sandstone specimens used for this study were prepared according to the mixture preparation procedure for poorly cemented formation proposed by Hashemi et al. [11] to achieve homogeneous properties. The mixture was composed of natural silica sands with two different grain size ranges, namely, coarse grain sands with grain size between 0.425 mm and 1.4 mm, and fine grain sands with a grain size between 0.15 mm and 0.355 mm, Portland cement type II (specific gravity, Gs = 3.15 g/cm$^3$) and water. The components were thoroughly mixed together to achieve a homogeneous mixture for the specimens, and the time spent between pouring water into the dry mixture and compacting it into metal moulds was maintained to be within 30 min to avoid initial setting of the cement. Each specimen was prepared by compacting the mixture into three equal layers of 42 mm thickness. The bottom surface between the layers was scratched before the subsequent layer was compacted on top to ensure thorough interlocking between the successive layers. The mixture was not strong enough to bond sand grains in the early curing stage. Thus, the specimens were left in the mould for five days, and then removed from the mould and cured for another three days wrapped in a plastic film at a room temperature (18–23 °C) before testing. The opposite ends of the specimen are made perfectly parallel. Further details for specimen preparation can be found in Hashemi et al. [11].

## 4. Design of the CMD

The convergence measuring device (CMD) was designed based on the local deformation transducer (LDT) developed at the Institute of Industrial Science, University of Tokyo, Tokyo, Japan [2]. The LD is a simple and low-cost device that is capable of accurately measuring axial strains to $10^{-4}$%. The working principle of the CMD is such that when the borehole of the TWHC specimen is deformed under triaxial stress conditions, the move-

ment of the CMD legs triggers either increase or decrease in the gage strain depending on the applied stress paths. The measurements taken from the two strain gages are then averaged and converted to displacement measurement through a calibration process by co-relating the CMD leg displacement and the output from the strain gages. The strain gage currently in use has a resistance of 120 ohms with a grid dimension of 2.5 mm wide and 10 mm long (TML FLA-10-11). The two strain gages form a half Wheatstone bridge to enhance measurement accuracy and minimize the temperature effect.

Figure 1 shows the final CMD manufactured for testing. It consists of strain gage attached to either side of a V-shaped metal frame which are positioned right opposite each other as shown below.

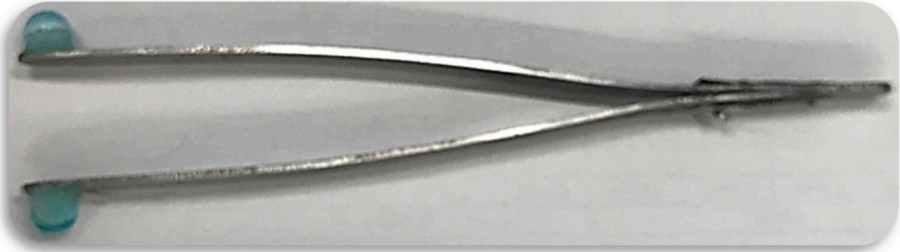

**Figure 1.** Convergence Measuring Device.

Each strain gage is placed near the closed end of the metal frame where it exhibits the strain in compression or tension when the open ends of the CMD legs move. To allow a smooth contact with the borehole wall, a circular plastic bead was attached to each side of the outer surface of the CMD legs using a strong glue.

## 5. Calibration of the CMD

The calibration of the CMD was carried out by comparing and modifying the output from the strain gage versus the displacement output from the digital calibrator. The initial distance between the open ends of the CMD was set at 15 mm, which matches the borehole diameter of the TWHC specimens. Both the CMD and the digital calibrator readings were zeroed for the initial measurement, and then subsequent measurements were taken at every 0.1 mm displacement as the distance between the CMD legs was contracted by 0.1 mm at a step to its full range capacity. A digital micrometre (Mitutoyo digimatic micrometre 164) with a resolution of 0.01 mm was used to get reference measurements for the calibration of the CMD. The calibration was carried out by comparing the measurements from the CMD and the digital micrometre output versus time.

## 6. Experimental Devices and Calibration

A triaxial stress system (MTS) was used for the test. This system is capable of supplying, maintaining, monitoring and controlling confining pressure as well as measuring accurate radial and axial displacements. Hoek cell was used to maintain pre-defined confining pressure on the specimen. Hydrostatic loading was applied for each of the stress paths tested. In order to measure accurate volumetric strain measurement, two axial variable displacement transformers (LVDT) and strain gauges were used. For the UCS testing, the specimens were prepared in three different cement contents to study the effect of cementing agent. UCS tests were conducted on the specimens of 65.3 mm in diameter and 127 mm in height.

A hollow cylinder test is used to apply an axial load and an external pressure along the curved surface of the cylinder specimen that provides a ready method for studying the strength. The modified Hoek cell designed and manufactured by Hashemi and Melkoumian [20] consisting of two sets of cylindrical platens was used for conducting the TWHC tests. A micro camera was fitted inside the test cell to monitor and record the borehole wall in real time. The Hoek cell was synchronised with a pressure maintainer specifically tuned for applying a low confining pressure without leakage. The CMDs were placed inside the

TWHC specimen as shown in Figure 2a. A hollow cylindrical metal spacer made from hardened steel matching the diameter of the TWHC specimen was placed in between the specimen and the hollow platen to maintain the CMD in the vertical position during testing. A total of two CMDs were used for each test to improve the accuracy of the convergence measurement. The top CMD was placed alongside the camera and approximately 42 mm from the top end surface of the specimen, and the bottom CMD was placed on the direct opposite location of the top CMD.

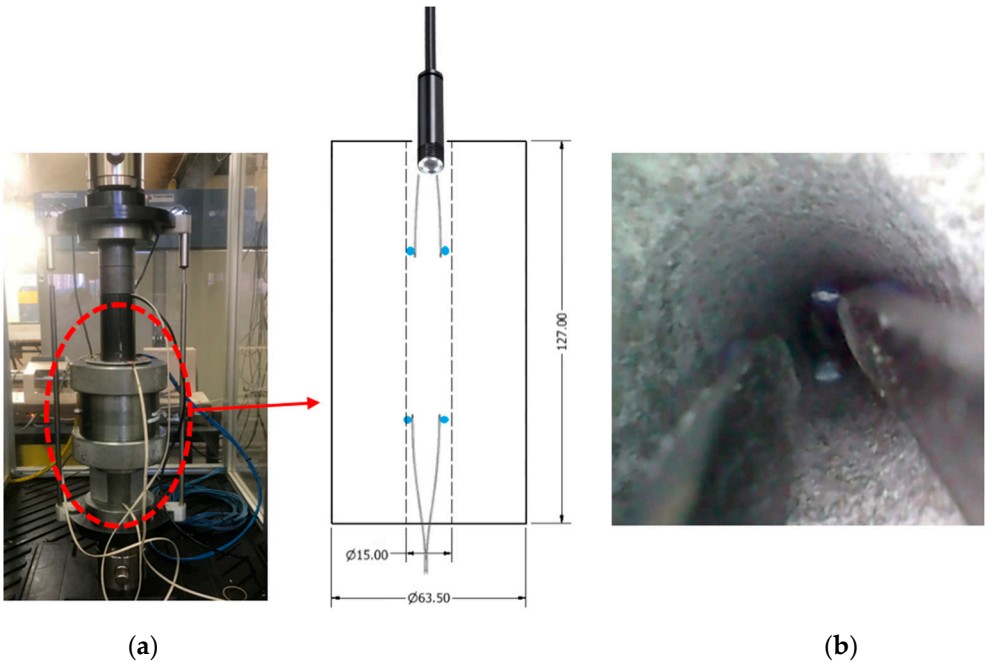

(**a**)                                                 (**b**)

**Figure 2.** (**a**) The CMDs placed inside the Hoek cell and (**b**) the view from the micro camera.

The end surfaces of each specimen were prepared to be flat and parallel to each other by applying a thin layer of dental paste. Lubrication was applied on the end surfaces in order to minimise the friction between the platen and the specimen. Furthermore, spherical platens were placed on top of the hollow platen to ensure that a uniform vertical stress was applied onto the TWHC specimen during testing and to minimise the bedding error. A pair of axial and lateral strain gages were attached directly to the specimen surface to measure local deformations. A small load of approximately 5 N was applied by the loading machine as the top ram was lowered down to the surface of the platen prior to the commencement of the test to ensure full contact between the top ram and the platen. The image captured by the micro camera as shown in Figure 2b was checked for the clear focal point of the camera and to ensure that the CMDs were positioned vertically. The experiments were performed with the following two loading stages: first, simultaneous vertical and confining stresses were applied up to a pre-defined stress level to simulate the hydrostatic stress condition in the first stage of loading. The second stage was followed by increasing the vertical load at a constant displacement rate of 0.07 mm/min. Confining pressure was varied in each test to apply desired confining stress to the TWHC specimen. Figure 3 illustrates borehole convergence observed by the camera placed above the specimen during a TWHC test under the first stress path without the CMD placed inside the borehole. The camera captured three different stages of borehole convergence; Figure 3 shows sand particles beginning to dislodge followed by visibly deformed borehole and then the fully converged borehole.

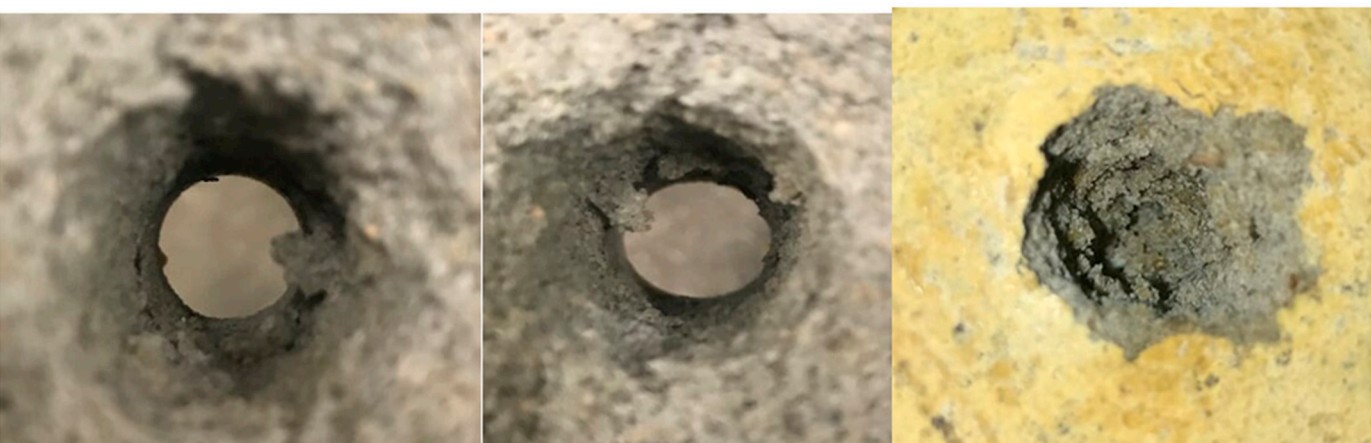

**Figure 3.** Borehole convergence stages observed by the camera.

### 7. Stress Paths

The effect of stress paths on the behaviour of TWHC was investigated by applying five different stress paths. Two approaches were used for analysing the stress paths: (1) considering the stresses on the boundary of the specimens and (2) considering certain stresses at the borehole wall. In the first approach (the first and second stress paths), the vertical stress ($\sigma_z$) and the confining pressure ($\sigma_{conf}$) were considered and the induced stresses at the borehole wall were calculated subsequently based on the applied stresses on the boundary of the TWHC specimens. In the second approach (the third, fourth and fifth stress paths), the principal stresses ($\sigma\theta$, $\sigma_r$, $\sigma_z$) for an element on the borehole wall were considered, and the stresses were applied to the boundaries of the specimen to induce the desired magnitude of tangential and vertical stresses on the borehole wall and on both ends of the specimen, respectively.

In the first stress paths (Equation (1)) both the vertical stress and the confining pressure were increased simultaneously at the same rate up to the pre-defined value, which simulated a hydrostatic condition acting on the boundary of the specimens. Then, in the second step, the specimen was subjected to a vertical loading increment (normal faulting) corresponding to a constant displacement rate of 0.07 mm/min. During the test, the level of the confining pressure applied to the external surface of the specimens was kept constant by an automatic pressure gauge system. In the second stress path when the stress level reached the hydrostatic condition, the confining pressure (reverse faulting) was increased, corresponding to a constant pressure rate of 0.2 MPa/min. The lateral displacement rate was measured by four 30 mm lateral strain gauges which were connected to the data acquisition and linked to the hydraulic pump systems.

In the third and fourth stress paths, $\sigma_\theta$ and $\sigma_z$ were increased simultaneously up to a certain value. Then, as per the experiment plan, the specimen was subjected either to $\sigma_z$ or $\sigma_\theta$ increment. In order to achieve the same tangential and vertical stresses at the borehole wall in the initial step of the test, the confining pressure value was derived in terms of the tangential stress, and the rate of increasing the vertical stress was kept constant and was the same as the rate of increase in the tangential stress at the borehole wall. It is worth mentioning that in an unsupported borehole, $\sigma_r$ at the borehole wall is zero.

In the fifth path, $\sigma_\theta$ and $\sigma z$ were increased simultaneously until the failure at the borehole wall was observed by the real-time camera recording. It should be mentioned that for each TWHC specimen, $\sigma_\theta$ and $\sigma z$ were calculated separately based on the stress application area and the borehole size. Equations (1)–(5) show the principal stress status at the boundary of specimens and borehole wall for each stress path. The difference between the first and fourth stress paths is in the initial step of the experiment. In the first path, the magnitude of $\sigma z$ is considerably lower than that for the fourth path until the end of the

first stage of the experiment. The same relationship was applied to the second and third stress paths.

$$\text{first stress path} \begin{cases} \sigma_1 = \sigma_2 = \sigma_3 = \sigma_{conf} = \sigma_z \ (\textit{first step}) \\ \sigma_1 = \sigma_z > \sigma_2 = \sigma_3 = \sigma_{conf} \ (\textit{second step}) \end{cases} \tag{1}$$

$$\text{second stress path} \begin{cases} \sigma_1 = \sigma_2 = \sigma_3 = \sigma_{conf} = \sigma_z \ (\textit{first step}) \\ \sigma_1 = \sigma_2 = \sigma_{conf} > \sigma_3 = \sigma_z \ (\textit{second step}) \end{cases} \tag{2}$$

$$\text{third stress path} \begin{cases} \sigma_1 = \sigma_2 = \sigma_\theta = \sigma_z > \sigma_3 = \sigma_r = 0 \ (\textit{first step}) \\ \sigma_1 = \sigma_\theta = \sigma_2 = \sigma_z > \sigma_r = 0 \ (\textit{second step}) \end{cases} \tag{3}$$

$$\text{fourth stress path} \begin{cases} \sigma_1 = \sigma_2 = \sigma_\theta = \sigma_z > \sigma_3 = \sigma_r = 0 \ (\textit{first step}) \\ \sigma_1 = \sigma_z > \sigma_2 = \sigma_\theta > \sigma_r = 0 \ (\textit{second step}) \end{cases} \tag{4}$$

$$\text{fifth stress path} \begin{cases} \sigma_1 = \sigma_\theta = \sigma_2 = \sigma_z > \sigma_r = 0 \ (\textit{first step}) \\ \sigma_1 = \sigma_\theta = \sigma_2 = \sigma_z > \sigma_r = 0 \ (\textit{second step}) \end{cases} \tag{5}$$

## 8. Results

To minimise the influence of rock heterogeneity on the test results, repetitive triaxial tests were performed on three specimens for each cement content under various confining pressures. Different values of confining pressure of 2 MPa, 4 MPa and 6 MPa.

### 8.1. First Stress Path

In the first stress path, both the axial stress and the confining pressure were increased simultaneously at the same rate up to the pre-defined level, in order to simulate hydrostatic condition on the boundary of the specimens. Then, in the second step, the specimen was subjected to an axial loading increment corresponding to a constant displacement rate of 0.07 mm/min. During the test, the level of the confining pressure applied to the external surface of the specimens was kept constant by an automatic pressure gauge system. The results showed the relatively linear function of the deviatoric stress followed by a constant increase in the radial strain as the borehole failure initiated. As shown in Figure 4a–c, this behaviour was more profound at a lower level of cement content of 10% compared to specimens with a higher cement content of 12% and 14%. Prior to the initiation of the borehole failure, the radial strain of the specimens with a higher amount of cement content increased less dramatically compared to that for the specimens with 10% cement content, indicating that the load-bearing capacity increases when the cement content is increased.

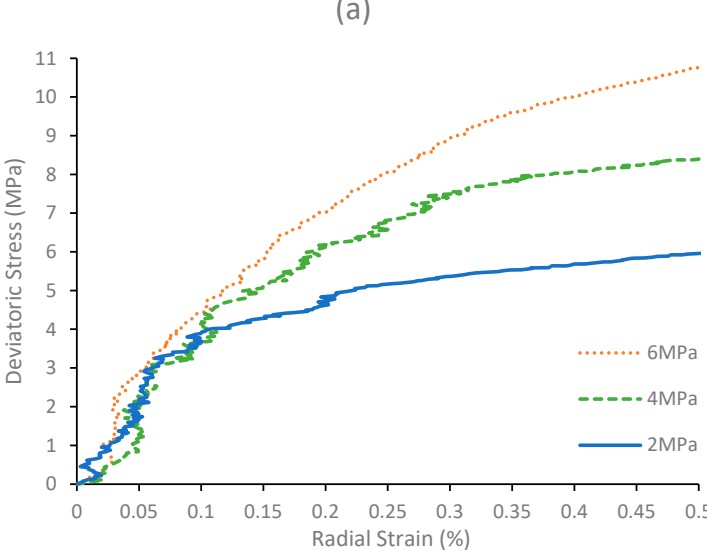

(a)

**Figure 4.** *Cont.*

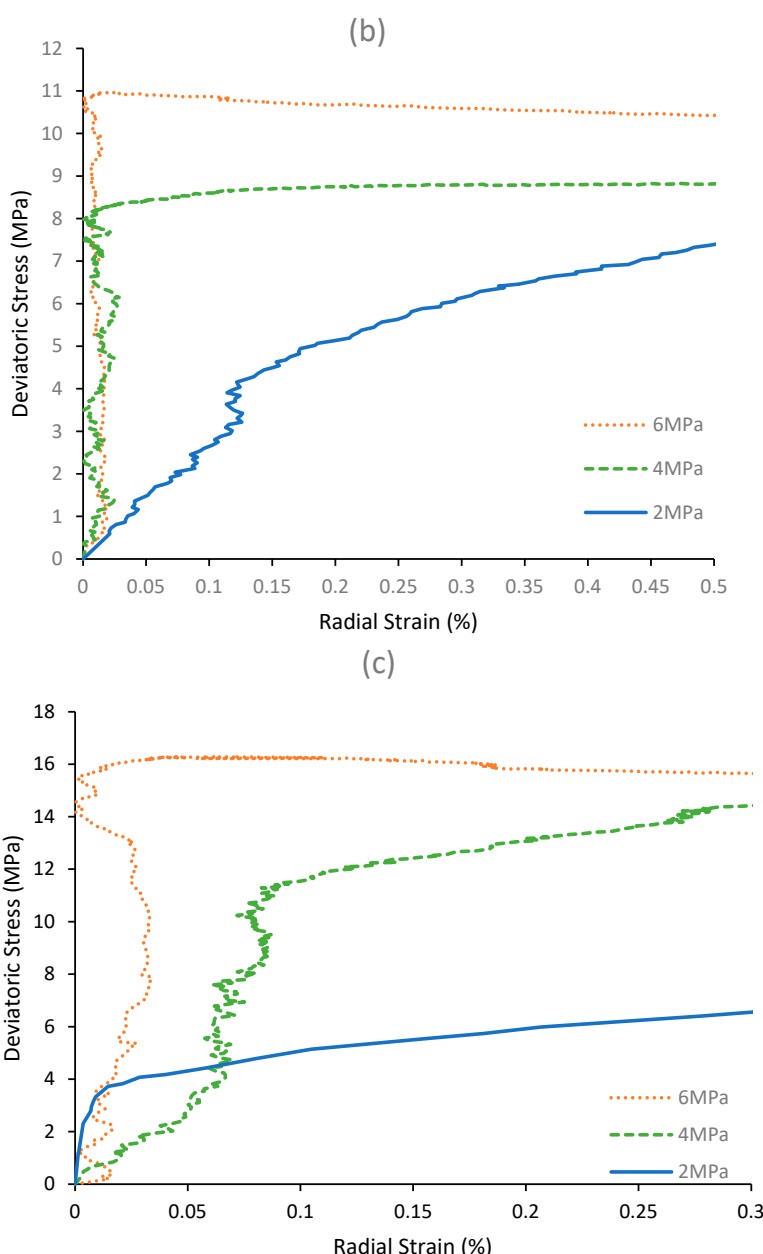

**Figure 4.** Deviatoric stress (MPa) versus radial strain (%): (**a**) 10% cement content, (**b**) 12% cement content, (**c**) 14% cement content.

### 8.2. Second Stress Path

The increase in the cement content had a minimal effect on the stiffness of the specimens tested under the second stress path. Mogi [21] suggested that the stiffness depends on the rock material and, as the same sand grain sizes were used for the specimen preparation, the specimen stiffness remained unchanged in the tests. The level of the confining pressure required to initiate a borehole convergence increased with the increase in the cement content. This is due to the strengthening effect of the cement. Hashemi et al. [11] have also observed that the poorly cemented specimens with a higher cement content demonstrate a higher peak strength due to the strengthening effect of the cementing agent. Generally, for the second path with an increase in the cement content, the radial strain increased for a certain confining stress. With increasing the cement content, the stiffness of the specimens in the vertical direction obtained higher values than in the lateral orientation for the second path; however, the rate of increase was lower for the second path.

Figure 5 shows that the effect of cement content has a higher influence on stabilizing a borehole than that of the confining stress applied for the second stress path. The level of confining pressure required to initiate a borehole convergence increased with the increase in the cement content. Such observation agrees with [11], where the poorly cemented specimens with a higher cement content demonstrated a higher peak strength due to the strengthening effect of the cementing agent. Figure 6 below illustrates the borehole failure process observed from the camera installed inside the borehole. Moreover, the specimens showed a more ductile behaviour when the equal amount of vertical stress and the maximum principal stress were applied to them. The borehole failure was observed at lower strains for the second path compared to the first path, which suggests that the tangential stress has the important influence on the borehole failure in poorly cemented specimens. Results displayed that the shear bands were formed, and the produced sand mainly consisted of uniform loose grains.

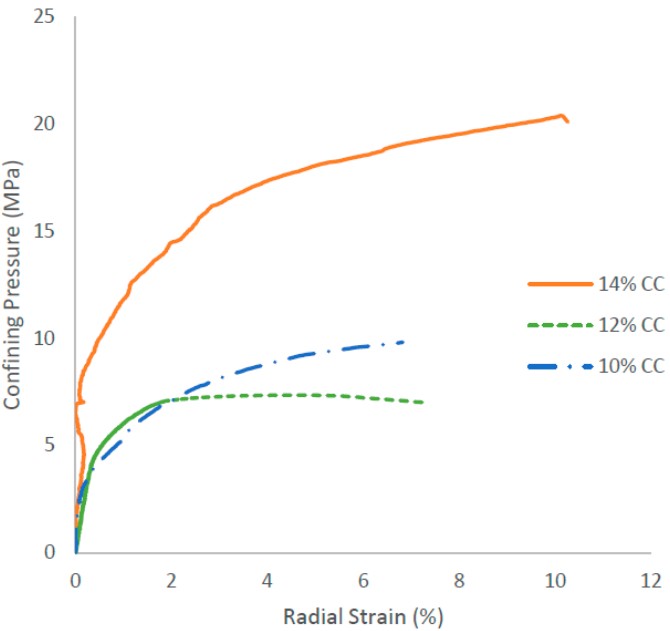

**Figure 5.** Confining pressure (MPa) versus radial strain (%).

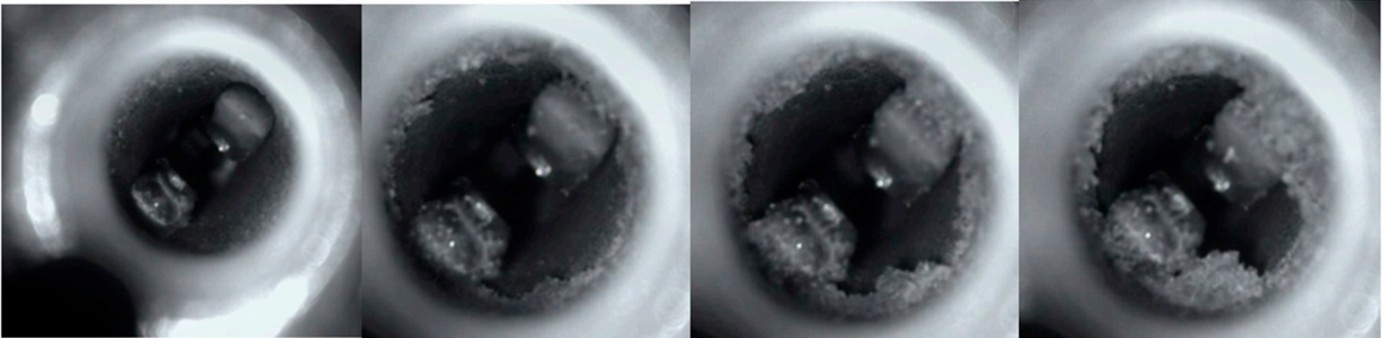

**Figure 6.** Borehole failure observations.

*8.3. Third Stress Path*

Figure 7a–c present the results on applying the third stress path on the TWHC specimens with 10%, 12% and 14% cement contents under 2 MPa, 4 MPa and 6 MPa, respectively. The confining and vertical stresses were considered for an element at the borehole wall based on the lateral and axial stresses applied on the boundary of the TWHC specimens. The stresses were applied to the specimen at the same rate until a given value was reached. Then the confining stress was increased until an instability at the borehole wall was ob-

served, while the vertical stress was kept unchanged. The main difference between the second and third stress paths was in the initial stage of the test, in which the axial stress applied to the specimen was considerably higher in the third path for the corresponding borehole size and specimen outer diameter. According to Figure 7a, with an increase in the vertical stress, the strength of the specimen reached prior to the initiation of the borehole failure was noticeably higher on the specimen with 14% cement under 6 MPa than the specimen that was under 4 MPa and 12% of cement content. It illustrates that the effect of vertical stress as a supporting pressure in increasing the ductility for a given cement content and the specimens exhibited a higher ductile behaviour when a greater vertical stress was applied. Furthermore, with increasing cement content, the borehole failure at the wall was observed at higher radial strains.

Unlike the specimens under the first stress path, the trends of graphs from the third stress path in general show that the vertical stress kept the specimen in the contraction mode and therefore, the dilation was observed at higher radial strains. The results also showed that with an increase in cement content, the radial strain would be increased for certain vertical stress in this stress path, similar to that in the second stress path. This confirms that with an increase in the cement content, the lateral strain increases in all the stress paths. However, the effect of cement content had less influence than the change in vertical stress. Increasing cement content from 10% to 12% did not result in any significant variation in radial strain.

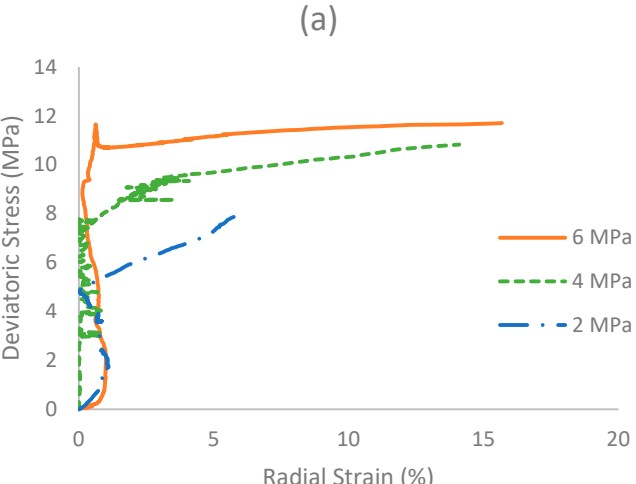

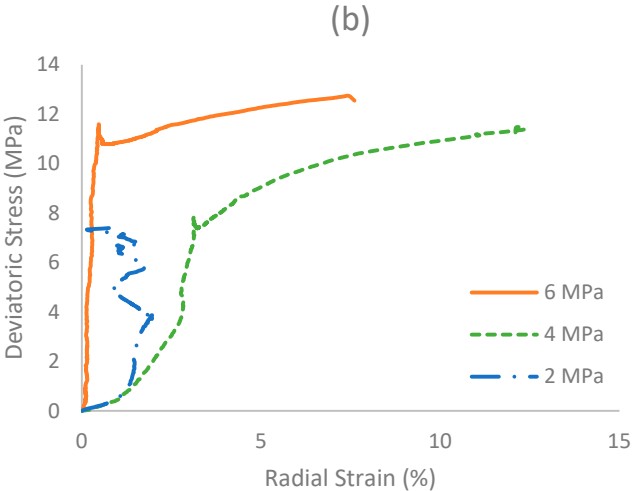

**Figure 7.** *Cont.*

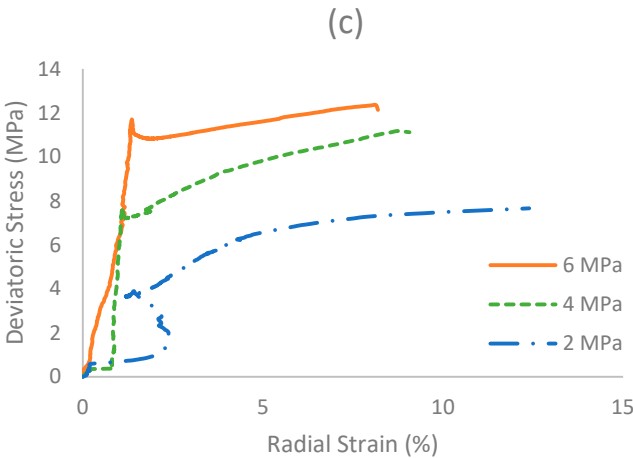

**Figure 7.** Deviatoric stress versus radial strain: (**a**) 10% cement content, (**b**) 12% cement content, (**c**) 14% cement content.

### 8.4. Fourth Stress Path

Figure 8a–c illustrate the stress–strain behaviour for the fourth stress path for 10%, 12% and 14% cement content respectively. In this path, the confining stress at the borehole wall was increased along with the axial stress at the same rate up to 2 MPa. Then, vertical stress was increased while confining stress was kept unchanged. The trend of the stress–strain relationships in the fourth path was similar to what was observed in the first path where the axial stress was much lower than that for the fourth path in the initial stage of the test. Figure 8a–c show that the borehole instability occurs at a higher stress and lower radial strain for the fourth path in different cement contents. Since the initial axial stress was higher for the fourth path, the specimens behaved in a brittle manner before the borehole convergence. Such behaviour is likely to be due to the effect of the loading rate on the strength of the tested specimens.

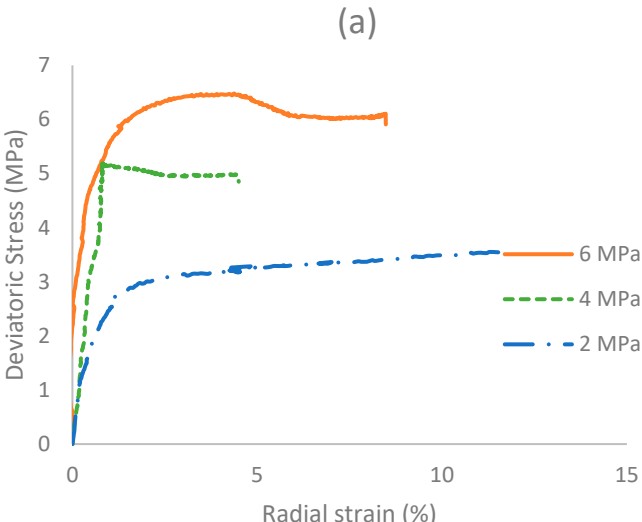

**Figure 8.** *Cont.*

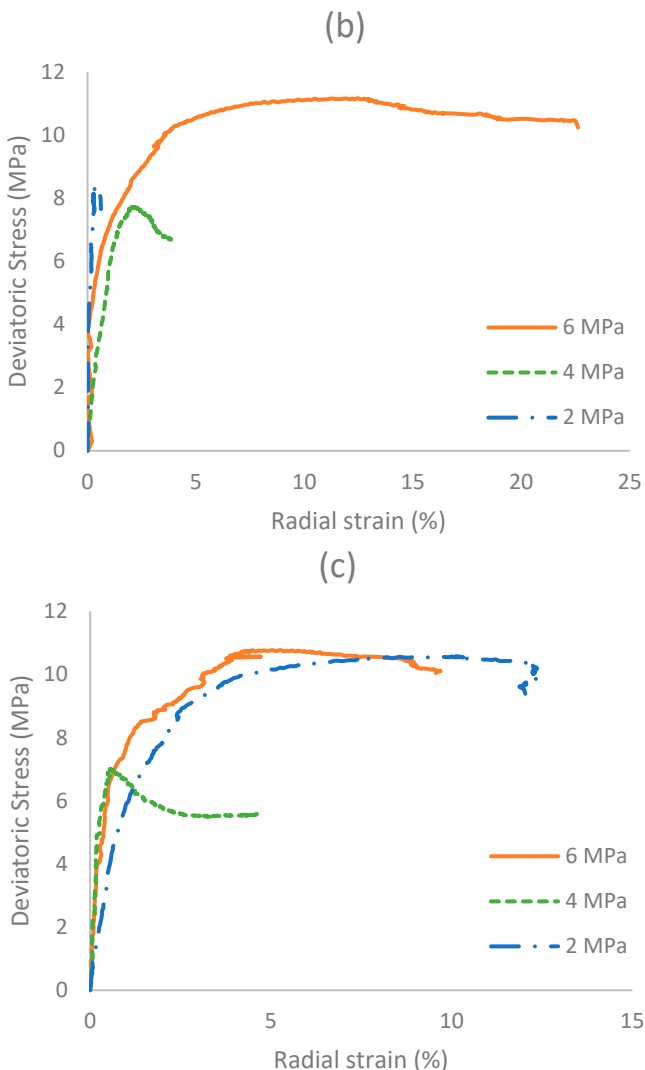

**Figure 8.** Stress versus radial strain: (**a**) 10% cement content, (**b**) 12% cement content, (**c**) 14% cement content.

*8.5. Fifth Stress Path*

In the fifth stress path, both $\sigma\theta$ and $\sigma r$ were increased simultaneously at the same rate until the failure at the borehole wall was observed by the camera. While the yield stresses increase consistently with the increase in the cement content, the specimens with 12% cement content behaved differently. For these specimens, the measured radial strain was significantly higher at a lower stress state, compared to the specimens with 10% and 14% of cement content. This is likely to be caused by one of the two CMDs being placed at a location in the borehole where the borehole converged significantly more than at the other CMD. Therefore, the averaging of the two CMD readings resulted in such a different reading compared to the results obtained for the other cement contents. The yield stress for specimen with 10% cement content was significantly less than the specimen with 14% cement content. The TWHC specimens with 10% and 14% cement content behaved similarly, as is indicated by the stress versus radial strain behaviour. When compared to that of the first and second stress paths where the confining pressure and axial stress were increased at a constant rate in the fifth stress path, the specimens tolerated a higher level of stresses surpassing 25 MPa, as can be seen in Figure 9. It is important to mention that the trend of the graph remained unchanged for different levels of stresses applied, demonstrating the consistent contraction mode under the fifth stress path.

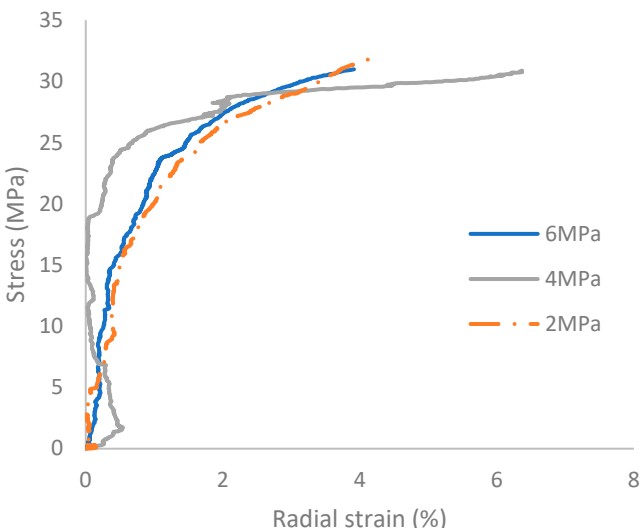

**Figure 9.** Deviatoric stress versus radial strain.

### 9. Conclusions

The effects of five different stress path regimes and three different cement agent contents on the borehole failure stress were investigated experimentally using the TWHC tests. Five different stress regimes were applied to the synthetic TWHC poorly cemented sand specimens with the CMDs deployed inside the borehole. In conjunction with the CMDs, the borehole was monitored by the camera. The five different stress paths included applying normal and strike–slip faulting stress regimes to synthetic TWHC poorly cemented sand specimens. The specimens under the first stress path showed that increase in the cement content increased the radial strain. The increase in the cement content had a minimal effect on the stiffness of the specimens tested under the second stress path. The specimens under third stress path showed that the radial strain increased with the increase in cement content. The effect of cement content had less influence on the borehole failure under the fourth stress path than the change in vertical stress. The specimens under the fifth stress path tolerated a higher level of stresses. Finally, the effect of the cement content on the borehole failure was found to be more significant than the effect of confining pressure in all stress paths.

**Author Contributions:** Conceptualization, J.H.H.; validation, J.H.H.; formal analysis, J.H.H.; investigation, J.H.H.; writing—original draft preparation, J.H.H.; writing—review and editing, J.H.H.; supervision, N.M. and S.S.H. All authors have read and agreed to the published version of the manuscript.

**Funding:** This research received no external funding.

**Conflicts of Interest:** The authors declare no conflict of interest.

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
