# Peer review of "Evaluating the Effect of Different Stress Path Regimes on Borehole Deformation Using Convergence Measuring Device"

_geosciences, doi:10.3390/geosciences12090317_

Round 1
Reviewer 1 Report
minor revision

Author Response
Response 1: Mechanical properties have been added in abstract Line 10
Response 2: Loading rate has been added in abstract Line 14
Response 3: the units and grammar have been checked and updated
Response 4: Fig 1 and 2b have been updated
Response 5: Conclusion on Page 14 has been updated to address Point 5
Response 6: Conclusion on Page 14 has been updated to address Point 6
Response 7: Suggested references have been added

Reviewer 2 Report
The review of the manuscript entitled: “Evaluating the Effect of Different Stress Path Regimes on Borehole Deformation using Convergence Measuring Device”. In this work, the authors applied five different stress regimes to the synthetic thick-walled hollow cylinder poorly cemented sand specimens with the convergence measuring device deployed inside the borehole. The work is very interesting, novel, and has a scientific style. Also, the work is within the journal's scope, and it is easy to follow. By responding to the following comments and questions, the work can be ready for publication:
1. Please improve the quality of some figures (for example fig. 1).
2. The main findings of obtained mechanism should be mentioned in the abstract.
3. It is recommended to extend the Introduction section by describing the cause, fatality, novelty of the work, advantages, and disadvantages of the present study.
4. In the “Introduction” section, it is recommended to compare the physical properties of carbonate and sandstone rocks during drilling and production. Various factors can affect their properties including permeability. In the revision stage, the following references can be used, in which rock permeability of carbonate samples was analyzed during production: “Petroleum Science and Technology, 36(18), 1482-1489”; “Petroleum Science and Technology, 36(14), 1030-1036”
5. Why were the tests completed at room temperature? How the real well conditions can be simulated through this?
6. The authors obtained that the lateral strain increases in all the stress paths with an increase in the cement content. The mechanism for this behavior should be added.
7. The conclusion also needs to be rewritten. Include the following: new concepts and innovations demonstrated in this study, summary of findings, comparison with findings by other workers, and concluding remark.
8. Use more recent references.
Author Response
Point 1: Please improve the quality of some figures (for example fig. 1).
Response 1: Fig 1 has been updated
Point 2: The main findings of obtained mechanism should be mentioned in the abstract.
Response 2: Additional findings have been added under line 19
Point 3: It is recommended to extend the Introduction section by describing the cause, fatality, novelty of the work, advantages, and disadvantages of the present study.
Response 3: Introduction has been updated
Point 4: In the “Introduction” section, it is recommended to compare the physical properties of carbonate and sandstone rocks during drilling and production. Various factors can affect their properties including permeability. In the revision stage, the following references can be used, in which rock permeability of carbonate samples was analyzed during production: “Petroleum Science and Technology, 36(18), 1482-1489”; “Petroleum Science and Technology, 36(14), 1030-1036”
Response 4: Both of the recommended references have been added
Point 5: Why were the tests completed at room temperature? How the real well conditions can be simulated through this?
Response 5: The tests have been completed at room temperature to replicate realistic environments of mineral exploration drillings carried out in Burra region of South Australia
Point 6: The authors obtained that the lateral strain increases in all the stress paths with an increase in the cement content. The mechanism for this behavior should be added.
Response 6: Conclusion has been updated
Point 7: The conclusion also needs to be rewritten. Include the following: new concepts and innovations demonstrated in this study, summary of findings, comparison with findings by other workers, and concluding remark.
Response 7: Conclusion has been rewriten
Point 8: Use more recent references.
Response 8: - A total of 4 recent references have been added including the references recommended by the reviewer

Reviewer 3 Report
Dear authors!
Your article is well structured, but it should be improved in some areas:
- Line 227: the figure 3 is stating 3 photographs but there is no explanation of which test these photos are taken from and what is the difference from photo to photo
- Line 229: the chapter 6. Stress paths should be explaining in more detail since the direction od the s1, s2, s3, sconf, sz, sr, sf is not explained and what each of the symbol is meaning
- Line 241: the 5 stress paths was proposed but there is no explanation why and how you decided that this combination of forces is the right one. Some more explanation is needed
- Line 269 - 276: you are giving the conclusion in the Results section what is not correct since all the conclusions should be given in the later chapter
- Line 282: you are showing only one graph a) (regarding the previous and next stress tests where is three graphs) but there is no explanation why this is so
- Line 297: the photos of the borehole failure is showing how the failure occurred but there are no photos for all 5 stress tests that would be more beneficial to the understanding of the borehole failure
- Line 369: chapter conclusion is not revealing much what you have concluded from your experiments, and it must be expanded
- The chapter Discussion is missing! Maybe the chapter Results is Results and discussion?
Best regards.
Author Response
Point 1: Line 227: the figure 3 is stating 3 photographs but there is no explanation of which test these photos are taken from and what is the difference from photo to photo
Response 1: Explanation for Fig 3 has been added under Line 233-235
Point 2: Line 229: the chapter 6. Stress paths should be explaining in more detail since the direction od the s1, s2, s3, sconf, sz, sr, sf is not explained and what each of the symbol is meaning
Response 2: Chapter 6 has been updated to explain stress paths in more details
Point 3: Line 241: the 5 stress paths was proposed but there is no explanation why and how you decided that this combination of forces is the right one. Some more explanation is needed
Response 3: Further explanation has been added
Point 4: Line 269 - 276: you are giving the conclusion in the Results section what is not correct since all the conclusions should be given in the later chapter
Response 4: Updated the commented section and the conclusion
Point 5: Line 282: you are showing only one graph a) (regarding the previous and next stress tests where is three graphs) but there is no explanation why this is so -
Response 5: Section 6. Stress Paths have been updated further and graph legend indicates different cement contents rather than confining pressures in the graphs under other stress paths.
Point 6: Line 297: the photos of the borehole failure is showing how the failure occurred but there are no photos for all 5 stress tests that would be more beneficial to the understanding of the borehole failure
Response 6: Acknowledged but the borehole convergence photos captured for different stress paths are similar visually, hence borehole convergence measuring device was used to measure the difference.
Point 7: Line 369: chapter conclusion is not revealing much what you have concluded from your experiments, and it must be expanded
Response 7: Conclusion has been expanded and re-written.
Point 8: The chapter Discussion is missing! Maybe the chapter Results is Results and discussion? -
Response 8: Results include both results and discussion
